# Development and validation of the CHIME simulation model to assess lifetime health outcomes of prediabetes and type 2 diabetes in Chinese populations: A modeling study

**Jianchao Quan[1], Carmen S. Ng[1]\*, Harley H. Y. Kwok[1], Ada Zhang[2], Yuet H. Yuen[1], Cheung-Hei Choi[3], Shing-Chung Siu[4], Simon Y. Tang[5], Nelson M. Wat[6], Jean Woo[7], Karen Eggleston[2,8], Gabriel M. Leung[1,9]**

**1** School of Public Health, LKS Faculty of Medicine, The University of Hong Kong, Hong Kong SAR, China, **2** Stanford University, Stanford, California, United States of America, **3** Queen Elizabeth Hospital, Hong Kong, China, **4** Department of Medicine & Rehabilitation, Tung Wah Eastern Hospital, Hong Kong, China, **5** Tuen Mun Hospital, Hong Kong SAR, China, **6** Caritas Medical Centre, Hong Kong SAR, China, **7** Faculty of Medicine, The Chinese University of Hong Kong, Hong Kong SAR, China, **8** National Bureau of Economic Research, Cambridge, Massachusetts, United States of America, **9** Laboratory of Data Discovery for Health, Hong Kong Science Park, Hong Kong SAR, China

\* csng14@hku.hk

**Data Availability Statement:** The data underlying the results presented in the study are available from the Hong Kong Hospital Authority Data

## Abstract

### Background

Existing predictive outcomes models for type 2 diabetes developed and validated in historical European populations may not be applicable for East Asian populations due to differences in the epidemiology and complications. Despite the continuum of risk across the spectrum of risk factor values, existing models are typically limited to diabetes alone and ignore the progression from prediabetes to diabetes. The objective of this study is to develop and externally validate a patient-level simulation model for prediabetes and type 2 diabetes in the East Asian population for predicting lifetime health outcomes.

### Methods and findings

We developed a health outcomes model from a population-based cohort of individuals with prediabetes or type 2 diabetes: Hong Kong Clinical Management System (CMS, 97,628 participants) from 2006 to 2017. The Chinese Hong Kong Integrated Modeling and Evaluation (CHIME) simulation model comprises of 13 risk equations to predict mortality, micro- and macrovascular complications, and development of diabetes. Risk equations were derived using parametric proportional hazard models. External validation of the CHIME model was assessed in the China Health and Retirement Longitudinal Study (CHARLS, 4,567 participants) from 2011 to 2018 for mortality, ischemic heart disease, cerebrovascular disease, renal failure, cataract, and development of diabetes; and against 80 observed endpoints from 9 published trials using 100,000 simulated individuals per trial. The CHIME model was compared to United Kingdom Prospective Diabetes Study Outcomes Model 2 (UKPDS-OM2) and Risk Equations for Complications Of type 2 Diabetes (RECODe) by assessing

Sharing Portal (https://www3.ha.org.hk/data/DCL/Index/), and the China Center for Economic Research, Peking University (https://charls.pku.edu.cn).

**Funding:** JQ received funding from Research Grants Council (ref. 27112518), Hong Kong SAR, China (https://www.ugc.edu.hk/eng/rgc). The funders had no role in study design, data collection and analysis, decision to publish, or preparation of the manuscript.

**Competing interests:** The authors have declared that no competing interests exist.

**Abbreviations:** ACCORD, Action to Control Cardiovascular Risk in Diabetes; ACE, Acarbose Cardiovascular Evaluation; ADVANCE, Action in Diabetes and Vascular disease: preterAx and diamicroN-MR Controlled Evaluation; AIC, Akaike's information criterion; BMI, body mass index; CDQDPS, China Da Qing Diabetes Prevention Study; CHARLS, China Health and Retirement Longitudinal Study; CHIME, Chinese Hong Kong Integrated Modeling and Evaluation; CMS, Clinical Management System; DPP, Diabetes Prevention Program; eGFR, estimated glomerular filtration rate; HDL, high-density lipoprotein; ICD-9-CM, International Classification of Disease, ninth revision; ICPC-2, International Classification of Primary Care, second edition; JDCS, Japan Diabetes Complications Study; J-DOIT3, Japan Diabetes Optimal Treatment study for 3 major risk factors of cardiovascular diseases; J-EDIT, Japan Elderly Diabetes Intervention Trial; JPAD, Japanese Primary Prevention of Atherosclerosis with Aspirin for Diabetes; LDL, low-density lipoprotein; NHS, National Health Service; QALY, quality-adjusted life year; RECODe, Risk Equations for Complications Of type 2 Diabetes; RMSE, root mean square error; RMSPE, root mean square percentage error; TRIPOD, Transparent Reporting of a multivariable prediction model for Individual Prognosis Or Diagnosis; UKPDS-OM2, UK Prospective Diabetes Study Outcomes Model 2; UKPDS 33 and 80, UK Prospective Diabetes Study 33 and 80.

model discrimination (C-statistics), calibration slope/intercept, root mean square percentage error (RMSPE), and $R^2$. CHIME risk equations had C-statistics for discrimination from 0.636 to 0.813 internally and 0.702 to 0.770 externally for diabetes participants. Calibration slopes between deciles of expected and observed risk in CMS ranged from 0.680 to 1.333 for mortality, myocardial infarction, ischemic heart disease, retinopathy, neuropathy, ulcer of the skin, cataract, renal failure, and heart failure; 0.591 for peripheral vascular disease; 1.599 for cerebrovascular disease; and 2.247 for amputation; and in CHARLS outcomes from 0.709 to 1.035. CHIME had better discrimination and calibration than UKPDS-OM2 in CMS (C-statistics 0.548 to 0.772, slopes 0.130 to 3.846) and CHARLS (C-statistics 0.514 to 0.750, slopes −0.589 to 11.411); and small improvements in discrimination and better calibration than RECODe in CMS (C-statistics 0.615 to 0.793, slopes 0.138 to 1.514). Predictive error was smaller for CHIME in CMS (RSMPE 3.53% versus 10.82% for UKPDS-OM2 and 11.16% for RECODe) and CHARLS (RSMPE 4.49% versus 14.80% for UKPDS-OM2). Calibration performance of CHIME was generally better for trials with Asian participants (RMSPE 0.48% to 3.66%) than for non-Asian trials (RMPSE 0.81% to 8.50%). Main limitations include the limited number of outcomes recorded in the CHARLS cohort, and the generalizability of simulated cohorts derived from trial participants.

## Conclusions

Our study shows that the CHIME model is a new validated tool for predicting progression of diabetes and its outcomes, particularly among Chinese and East Asian populations that has been lacking thus far. The CHIME model can be used by health service planners and policy makers to develop population-level strategies, for example, setting HbA1c and lipid targets, to optimize health outcomes.

## Author summary

### Why was this study done?

- The chronic progression to diabetes-related complications is suitable for computer simulation modeling due to the long-term nature of health outcomes and the time lag for interventions to impact upon patient outcomes.

- Existing predictive outcomes models for type 2 diabetes developed and validated in historical European populations may not be applicable for East Asian populations due to differences in epidemiology and complications.

- A validated tool to predict lifetime health outcomes for prediabetes and type 2 diabetes in the Chinese population is needed.

### What did the researchers do and find?

- We developed the Chinese Hong Kong Integrated Modeling and Evaluation (CHIME) simulation model as a validated tool for predicting progression of diabetes and related

outcomes in Chinese and East Asian populations using Clinical Management System (CMS) (2006 to 2017) and China Health and Retirement Longitudinal Study (CHARLS) (2011 to 2018).

- The CHIME outperformed the widely used United Kingdom Prospective Diabetes Study Outcomes Model 2 (UKPDS-OM2) and Risk Equations for Complications Of type 2 Diabetes (RECODe) models on real-world data.

- Validation of the CHIME model was more accurate for trials with mainly Asian participants than trials with mostly non-Asian participants.

### What do these findings mean?

- Our study showed that the CHIME model is a new validated tool for predicting outcomes in Chinese and East Asian populations with prediabetes and type 2 diabetes.

- Existing diabetes outcomes models developed in European or North American populations may not be applicable to Chinese populations.

- Diabetes outcomes models such as the CHIME model can be used by health service planners and policy makers to develop population-level strategies to optimize health outcomes.

## Introduction

China has by far the largest absolute burden of diabetes, with an estimated 116 million adults living with the disease accounting for one-quarter of patients with diabetes globally [1]. Diabetes-related health expenditure for China alone reached USD 109 billion [1]. Worryingly, the prevalence of prediabetes has risen to 35.7% of Chinese adults [2], and the diabetes epidemic is expected to increase to 147 million adults by 2045.

Evaluating the health and economic outcomes of diabetes and its complications is vital for formulating health policy. The chronic progression to diabetes-related complications is apt for computer simulation modeling due to the long-term nature of health outcomes and the time lag for interventions to impact upon patient outcomes. Yet differences in epidemiology and outcomes among East Asian populations with diabetes render application of existing models that were developed and validated in European and North American populations problematic [3]. The most widely used model, United Kingdom Prospective Diabetes Study Outcomes Model 2 (UKPDS-OM2), is underpinned by risk equations from a 1970s UK cohort and overestimates the absolute risks of coronary heart disease and stroke among East Asians [4,5]. The more recent Risk Equations for Complications Of type 2 Diabetes (RECODe) model for 10-year risks was developed from a trial in the United States/Canada and has been validated in both North American trials and cohorts [6,7]. Other existing diabetes models such as CDC-RTI, CORE, and BRAVO were all developed from trials conducted in European or North American settings with few, if any, Asian participants and have rarely been tested by external validation on individual-level data [8–10] (see S1 Table).

We sought to develop and validate an outcomes model for the development of diabetes and related complications derived from Chinese (East Asian) populations and compare this new

Chinese Hong Kong Integrated Modeling and Evaluation (CHIME) model to the existing UKPDS-OM2 and RECODe models. Despite the continuum of risk across the spectrum of risk factor values, existing models are typically limited to diabetes alone and ignore the progression from prediabetes to diabetes. The CHIME simulation model integrates prediabetes and diabetes into a comprehensive outcomes model comprising of 13 outcomes including mortality, micro- and macrovascular complications, and development of diabetes. The lack of an appropriate simulation model for East Asia and prediabetes is a major gap for economic evaluation of interventions. The CHIME model can be applied as a tool to assist clinical and policy decision-makers evaluate management strategies over the lifetime horizon.

## Methods

The analyses per se were not prespecified but have formed part of a multinational research project studying the long-term costs of diabetes care. As part of that work, we planned to undertake risk prediction modeling to assess the net value of medical spending on diabetes care using longitudinal patient-level from multiple health systems in Asia, Europe, and North America [11,12].

The analyses for model development were planned after obtaining and reviewing the Hong Kong Hospital Authority Clinical Management System (CMS) data, without which we did not a priori understand or had access to even the data fields and structures available. We planned to validate against simulated cohorts from 9 trials determined in advance from validation studies of existing diabetes outcome simulation models and diabetes trials conducted in East Asia (S1 Table). We subsequently obtained individual-level data for model validation from the China Health and Retirement Longitudinal Study (CHARLS) cohort. We planned to compare model performance of CHIME with the existing UKPDS OM2 model. Further comparison with the recently developed RECODe model and calibration assessments by slope and intercept were made in response to peer review. The analyses used only deidentified data, and the study was approved by the respective institutional review board for each Hospital Authority cluster (Hong Kong East/West, Kowloon Central/East/West, New Territories East/West).

The CHIME model was developed using CMS data and externally validated against CHARLS cohort and 9 published trials. CMS is one of the largest Chinese electronic health informatics systems with detailed clinical records. CHARLS was chosen for external validation as it is a nationally representative longitudinal cohort of middle-aged and elderly Chinese residents age 45 and older. We validated against 6 outcomes measures recorded in the CHARLS data and an additional 80 endpoints from 9 published trials of diabetes patients using simulated cohorts of 100,000 individuals.

### Study populations

**CMS.** Hong Kong has a population of 7.5 million (92% Chinese) [13]. The estimated prevalence of prediabetes and diabetes in Hong Kong was 8.9% and 10.3%, respectively, in 2014 [14]. In Hong Kong, universal public healthcare is provided by Hospital Authority—a statutory body modeled after the British National Health Service (NHS) that manages public hospitals and ambulatory clinics. The Hospital Authority system provides care for 95% of people with diabetes in Hong Kong [15].

The Hospital Authority CMS is the health informatics system for the publicly provided healthcare in Hong Kong [16]. Electronic health records in CMS are linked via unique patient identity numbers and include patient demographics, records of deaths, admissions, attendances, diagnoses, procedures, medications, and laboratory tests. Diagnoses are coded according to the International Classification of Disease, ninth revision (ICD-9-CM) and the International Classification of Primary Care, second edition (ICPC-2).

We included adults diagnosed with either prediabetes or type 2 diabetes from January 1, 2006 to December 31, 2017. We defined prediabetes based on American Diabetes Association criteria, namely HbA1c 5.7% to 6.4% (39 to 47 mmol/mol), fasting glucose 5.6 to 7.0 mmol/L (100 to 125 mg/dL), or oral glucose tolerance test 7.8 to 11 mmol/L (140 to 199 mg/dL) [17]. We defined type 2 diabetes according to an algorithm for electronic healthcare records established in previous studies on the Hong Kong dataset [14], namely HbA1c ≥6.5% (≥48 mmol/mol); fasting plasma glucose ≥7.0 mmol/L (≥126 mg/dL); oral glucose tolerance test ≥11.1 mmol/L (200 mg/dL); random plasma glucose ≥11.1 mmol/L (≥200 mg/dL) on 2 separate occasions; diagnosis code for diabetes; or prescription of antihyperglycemic medication. We excluded individuals under the age of 20 at the date of onset of diabetes or prediabetes (whichever is earlier) or with a diagnosis code for type 1 diabetes.

We included 13 outcomes in our model development: all-cause mortality, diabetes-related macrovascular events (myocardial infarction, ischemic heart disease, heart failure, and cerebrovascular disease), microvascular events (peripheral vascular disease, neuropathy, amputation, ulcer of the skin, renal failure, cataracts, and retinopathy), and development of diabetes status. Clinical outcomes were extracted using diagnostic codes from the CMS dataset and mortality records from the Hong Kong death registry (detailed definitions of clinical outcomes with specified diagnostic codes shown in S2 Table).

**CHARLS.** The overall prevalence of diabetes and prediabetes in mainland China was 12.8% and 35.2% in 2018 [18]. CHARLS is a nationally representative longitudinal cohort of Chinese residents ages 45 and older. Details of the CHARLS cohort profile and biomarkers have been previously published [19,20]. The baseline survey wave was conducted between June 2011 and March 2012 and included 10,000 households in 150 counties/districts and 450 villages/resident committees using multistage stratified probability sampling. The CHARLS survey excluded individuals from collective dwellings such as school dormitories, nursing homes, and military bases. The response rate of the baseline wave was 80.5% [19]. All individuals were followed up in survey waves 2 to 4 conducted in 2013, 2015, and 2018. We derived the sample of prediabetes and diabetes participants from the 2011 baseline wave in line with the case definition algorithm based on measured HbA1c, fasting serum glucose, and self-reported diabetes status previously applied by Zhao and colleagues [21]. The 6 outcomes used for validation in the CHARLS cohort were mortality, ischemic heart disease, cerebrovascular disease, renal failure, cataract, and diabetes status.

**Simulated trial cohorts.** To further validate the CHIME model against additional outcomes, we compared the predicted against observed rates of 80 endpoints from 9 published trials of diabetes and prediabetes with long-term follow-up data, defined as trial length greater than 4 years. There were 4 trials conducted in East Asia (1 Chinese and 3 Japanese) and 5 trials outside Asia: Acarbose Cardiovascular Evaluation (ACE) [22], Action to Control Cardiovascular Risk in Diabetes (ACCORD) [23], Action in Diabetes and Vascular disease: preterAx and diamicroN-MR Controlled Evaluation (ADVANCE) [24], Diabetes Prevention Program (DPP) [25], Japan Diabetes Complications Study (JDCS) [26], Japan Elderly Diabetes Intervention Trial (J-EDIT) [27], Japanese Primary Prevention of Atherosclerosis with Aspirin for Diabetes trial (JPAD) [28], and UK Prospective Diabetes Study 33 and 80 (UKPDS 33 and 80) [29,30].

## Predictors

In order to predict future health outcomes, we selected candidate predictors from a review of existing diabetes outcomes models in Mount Hood Diabetes registry of simulation models [31] (see S1 Table) and input from clinician experts within the authorship group. They included age, sex, diabetes status, duration of diabetes, smoking status, body mass index

(BMI), glycated hemoglobin (HbA1c), systolic blood pressure, diastolic blood pressure, high-density lipoprotein (HDL) cholesterol, low-density lipoprotein (LDL) cholesterol, triglycerides, estimated glomerular filtration rate (eGFR), hemoglobin, and white blood cell count; medications (insulin, non-insulin hypoglycemic agent, antihypertensives, and statins); and preexisting medical conditions (atrial fibrillation, myocardial infarction, ischemic heart disease, heart failure, cerebrovascular diseases, peripheral vascular diseases, neuropathy, amputation, renal failure, hemodialysis, retinopathy, cataract, and ulcer of skin). Since the CMS dataset accounts for more than 90% of total bed days in the Hong Kong healthcare system during the study period [32], we assumed that the health outcome data were essentially complete and therefore missing data in the predictor variables was not dependent on the outcome. Complete case analysis was preferred over multiple imputation as only the predictors have missing values and the probability to be missing does not depend on outcome [33–35]. Individuals in the derivation cohort with missing predictor data at baseline are shown in S3 Table.

## Statistical methods

We used parametric proportional hazard models to analyze our data by fitting multivariable models incorporating time-varying clinical biomarkers and comorbidities for each outcome, in which time since enrollment was employed as the time interval. The model fitting process for the final risk models were based on a combination of backwards selection process using Akaike's information criterion (AIC) and consultation from clinical experts within the authorship group. Selected variables were assessed for clinical relevance with the outcome and the direction of associations with final selection based on group consensus among the clinical experts. The parametric form of the underlying hazard was examined graphically, and models were selected by AIC for exponential, log-logistic, log-normal, and Weibull parametric distributions, where lower AIC was considered to indicate a better model fit. We applied moving averages to smooth fluctuation for biomarkers by averaging the parameters values for each year. Continuous variables were modeled as nonlinear using restricted cubic spline function with 3 knot points at 10%, 50%, and 90% percentiles [36]. We also included the history of previous events, so an event occurring at baseline or during the previous model cycle would be recorded as a history of that event for the current yearly cycle. For internal validation, we calculated the overfitting bias corrected Harrell's C-statistic and Brier score at 10 years using bootstrap resampling with 100 replications [36]. Harrell's C-statistic is an extension of the receiver operating characteristic statistic for survival data. Brier score was based on the predicted and observed cumulative incidence at 10 years.

A schematic of the CHIME model structure is illustrated in Fig 1. The risk equations were applied annually in an individual-level discrete-time simulation model. Model inputs were entered for each individual including their baseline demographics, clinical risk factors, and history of complications. The simulation involved using the risk equations to estimate the probability of each outcome for each individual to determine whether the event occurred or not during the annual cycle. If the simulation predicted that an individual died in that annual cycle, the time to death and time to other outcomes were recorded. If the individual survived the annual cycle, their age, duration, history of events, and risk factor values were updated for entry into the next cycle. Thus, the individual's updated risk factors and history of events are used to predict the occurrence of outcomes and changes in risk factors in the next annual cycle. The discrete-time cycles are then repeated sequentially for the length of the simulation time. The simulation model recorded outputs including time to death and complications, annual incidence of complications and death, and changes in risk factors. For individuals with prediabetes, the simulation also recorded time to progression to diabetes.

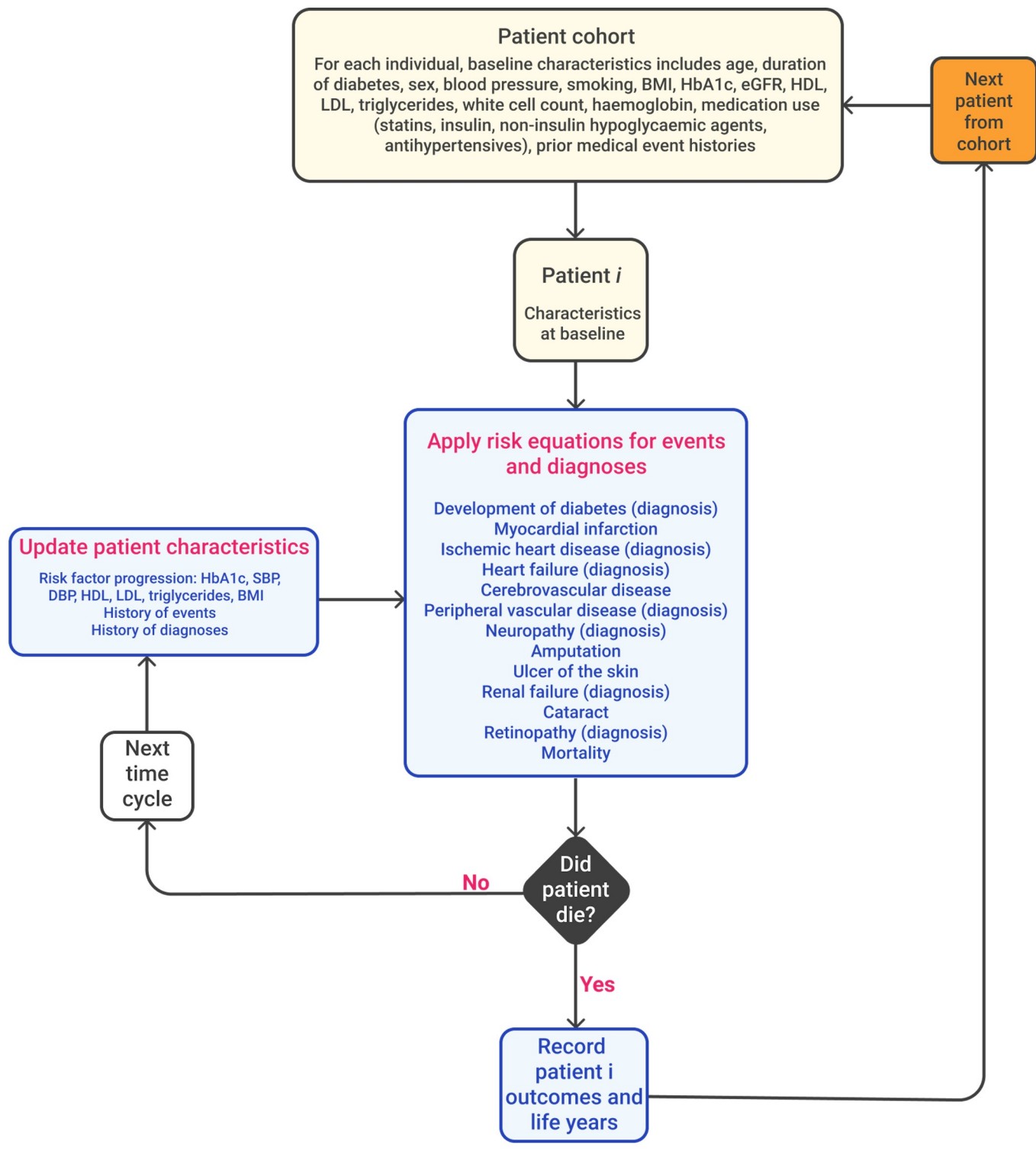

Each risk equation is applied to patient *i* from simulated cohort (diabetes risk equation is applied if patient *i* is pre-diabetic, all-cause mortality is last) for every year until time horizon is reached or until patient has died. If patient has not died or reached time horizon, simulation will update patient age, duration of diabetes, and event history for the next year. If patient has reached time horizon or died, simulation will move on to next patient in the cohort.

**Fig 1. Schematic of CHIME simulation model structure.** CHIME, Chinese Hong Kong Integrated Modeling and Evaluation; eGFR, estimated glomerular filtration rate; HDL, high-density lipoprotein; IHD, ischemic heart disease; LDL, low-density lipoprotein; MI, myocardial infarction; PVD, peripheral vascular disease.

We predicted the progression of risk factor values (glycated hemoglobin HbA1c, systolic blood pressure, diastolic blood pressure, HDL cholesterol, LDL cholesterol, triglycerides, and BMI) upon completion of each discrete-time cycle. To do this, we modeled the trajectory for each biomarker (continuous variables) over time for the study participants in the CMS dataset using ordinary least squares regression. The biomarker value for each risk factor in the current cycle was predicted by its lagged average values in the previous 2 years, age, sex, duration, and medications (statins, hypoglycemic agents, and antihypertensives), in keeping with other diabetes outcomes models [10]. Model fit was assessed by the root mean square error (RMSE) as the variables are continuous.

## Validation

**Comparisons to other outcomes models.** We identified previous diabetes outcome models registered in Mount Hood Diabetes registry of simulation models [31]; participants characteristics, model development, and validation strategies are detailed in S1 Table. Almost all identified models were proprietary and did not have publicly available code; only the UKPDS-OM2 and RECODe had user interfaces for comparative performance assessment. The UKPDS equations take various functional forms specific to each outcome (e.g., Gompertz, Weibull, logistic, or exponential), whereas the RECODe equations are Cox proportional hazards models [5,6].

The RECODe model predicts risks at a specified period of 10 years; for comparison, we assessed the CHIME, UKPDS-OM2, and RECODe models against CMS participants enrolled from 2006 to 2008 and followed until December 31, 2017. We compared the CHIME and UKPDS-OM2 models against the CHARLS validation cohort at 6 years of follow-up (wave 4 conducted in July to September 2018). Since UKPDS-OM2 and RECODe do not predict for participants with prediabetes, for our main analyses, we compared all models against CMS and CHARLS participants with type 2 diabetes only (heart rate data were unavailable for CHARLS participants).

We assessed model discrimination using the C-statistic at 10 years for CMS and 6 years for CHARLS with confidence intervals estimated from 100 bootstrap replications. We assessed calibration through the slope and intercept of the line between predicted and observed probabilities of each outcome by deciles of risk, with fewer centiles than deciles used if fewer than 5 events were observed per group to prevent unstable inferences [37]. We also measured the goodness of fit between predicted and observed endpoints using the root mean square percentage error (RMSPE), where lower scores indicate better fit, and present scatterplots of predicted versus observed endpoints along with the coefficient of determination ($R^2$).

**Simulation against published trial.** Since the CHARLS data only reported mortality, ischemic heart disease, cerebrovascular disease, renal failure, cataracts, and diabetes status, we also performed validation against published trial data, in keeping with the performance assessment strategy used in most diabetes outcomes models that lacked an individual-level validation cohort (see S1 Table for further details) [8–10]. We used the published baseline characteristics of the trial participants to generate a simulated cohort for the duration of each respective trial, with separate cohorts for each arm of the trial to model differing treatment effects between intervention and control arms.

We modeled the entire distribution of risk factors to account for sampling uncertainty, patient heterogeneity, and prior history when extrapolating clinical trial data [38]. For each

individual participant at baseline, we took the reported mean and standard deviation for each continuous variable (e.g., age, duration of diabetes, and biomarkers) to randomly generate values assuming a normal distribution. We used rnorm function (R version 3.6.3) to generate the random values. Upper and lower bounds for generated values were set according to the inclusion and exclusion criteria in the study protocol for each trial. For example, in the treatment arm of the ACE trial, the age value of each participant used a normal distribution centered around mean 64.4 years with a standard deviation of 8.2 years, truncated at a lower limit of 50 years old (inclusion criterion). For binary and categorical variables (sex, smoking status, prescribed medications, and past medical history), we took the percentage of participants with the particular status as the sampling probability. For example, in the treatment arm of the ACE trial, the sampling probability for female was 27%.

To simulate the trial progression, we assumed that the full treatment effects on each measured biomarker was reached in year 2 and remained stable thereafter for the remainder of the trial. Full treatment effect was defined as the maximal benefit and operationalized as a percentage of the average value at baseline. For example, in the treatment arm of the ACE trial, the full treatment effect on HbA1c was an average decrease of 0.05 percentage points from the average baseline HbA1c value of 5.9. Thus, each participant in the simulated cohort had a relative decrease of 0.05/5.9 or 0.85% from the first year HbA1c value in the subsequent years. We made no attempt to calibrate the model outputs to each individual trial. The point estimates for each predicted endpoint were obtained from simulating at least 100,000 participants per trial.

### Transparency and reporting

Statistical codes can be found on the GitHub repository: https://github.com/quan-group/CHIME. All analyses were carried out using R version 3.6 (R Foundation for Statistical Computing, Vienna, Austria) using the rms package [36]. This simulation model has been registered on the Mount Hood Diabetes Challenge Network, a registry includes a set of reference simulations that are intended to enable comparisons of models across time; for further details, see [15]. This study was approved by the Institutional Review Boards of all Hong Kong Hospital Authority clusters: HKWC, HKEC, KC/KEC, KWC, NTWC, and NTEC. This study follows the reporting guidelines in the Transparent Reporting of a multivariable prediction model for Individual Prognosis Or Diagnosis (TRIPOD) statement [39]; see S1 TRIPOD Checklist.

## Results

Baseline characteristics of the study participants in CMS and CHARLS cohorts are provided in Table 1. The CMS development cohort had 97,628 participants in the Hong Kong with type 2 diabetes (43.5%) or prediabetes (56.5%), with a mean follow-up time of 4.1 years (range of 0 to 12.8 years, accruing 402,250 person-years). The CHARLS validation cohort had 4,567 participants of which 216 (4.7%) were missing at 6 years follow-up (S4 Table). The CHARLS cohort was younger than the CMS cohort (mean 59.5 years versus 61.9 years), had lower HbA1c (5.5% versus 6.7%), lower BMI (23.9 versus 25.3), better renal function, and similar lipid profiles, but consisted of more smokers (28.4% versus 11.1%) and fewer people on medications.

We observed 9,878 deaths in CMS data during the follow-up period, equivalent to annual rate of 0.025 (the number of events for each outcome during the follow-up period is presented in S5 Table). The predictors included in the CHIME biomarkers and outcomes model are shown with coefficients and standard errors (S6 Table) and survival time ratios (S7 Table).

Table 2 shows the validation performance for CHIME, UKPDS-OM2, and RECODe against the CMS and CHARLS individual level datasets. In internal validations, CHIME C-statistics

**Table 1. Baseline characteristics of participants included in the study.**

| Characteristic | CMS development cohort | | | | | | CHARLS validation cohort | | | | | |
|---|---|---|---|---|---|---|---|---|---|---|---|---|
| | Prediabetes (n = 55,133) | | Diabetes (n = 42,495) | | Total (n = 97,628) | | Prediabetes (n = 3,361) | | Diabetes (n = 1,206) | | Total (n = 4,567) | |
| Age, years (Mean/SD) | 63.4 | 12.8 | 60.0 | 12.6 | 61.9 | 12.8 | 59.3 | 9.3 | 60.1 | 9.1 | 59.5 | 9.3 |
| Female (n/%) | 27,782 | 50.4 | 18,485 | 43.5 | 46,267 | 47.4 | 1814 | 54 | 660 | 54.7 | 2474 | 54.2 |
| Duration of diabetes, years (median/range) | - | - | 0 | 0.0–12.0 | 0 | 0.0–12.0 | - | - | 0 | 0.0–50.0 | 0 | 0.0–50.0 |
| Smoking status (n/%) | | | | | | | | | | | | |
| Current smoker | 5,031 | 9.1 | 5,827 | 13.7 | 10,858 | 11.1 | 954 | 28.4 | 341 | 28.3 | 1295 | 28.4 |
| Past smoker | 9,689 | 17.6 | 7,977 | 18.8 | 17,666 | 18.1 | 318 | 9.5 | 118 | 9.8 | 436 | 9.5 |
| Biomarkers (Mean/SD) | | | | | | | | | | | | |
| BMI (kg/m$^2$) | 25.1 | 4.1 | 25.6 | 4.3 | 25.3 | 4.2 | 23.7 | 3.8 | 24.6 | 3.9 | 23.9 | 3.8 |
| HbA1c (%) | 5.9 | 0.3 | 7.8 | 1.7 | 6.7 | 1.5 | 5.2 | 0.4 | 6.2 | 1.6 | 5.5 | 1.0 |
| Systolic blood pressure (mm Hg) | 133.9 | 14.3 | 135.3 | 15.4 | 134.5 | 14.8 | 129.3 | 18.5 | 132.7 | 18.7 | 130.2 | 18.6 |
| Diastolic blood pressure (mm Hg) | 76.5 | 10.0 | 77.6 | 9.7 | 77 | 9.9 | 75.5 | 11.3 | 76.6 | 11.0 | 75.8 | 11.3 |
| HDL cholesterol (mmol/L) | 1.4 | 0.4 | 1.3 | 0.3 | 1.3 | 0.4 | 1.3 | 0.4 | 1.2 | 0.4 | 1.3 | 0.4 |
| LDL cholesterol (mmol/L) | 3.0 | 0.8 | 3.0 | 0.8 | 3.0 | 0.8 | 3.1 | 0.9 | 3.0 | 1.0 | 3.1 | 1.0 |
| Triglycerides (mmol/L) | 1.4 | 0.7 | 1.6 | 0.9 | 1.5 | 0.8 | 1.5 | 1.0 | 2.1 | 1.7 | 1.6 | 1.2 |
| Hemoglobin (g/L) | 13.4 | 1.6 | 13.7 | 1.7 | 13.5 | 1.6 | 14.5 | 2.1 | 14.5 | 2.2 | 14.5 | 2.1 |
| White cell count (×10$^9$) | 7.3 | 2.1 | 8.0 | 2.3 | 7.6 | 2.2 | 6.3 | 1.9 | 6.4 | 1.9 | 6.4 | 1.9 |
| eGFR (mL/min/1.73m$^2$) | 86.8 | 23.2 | 92.2 | 28.4 | 89.2 | 25.7 | 104.5 | 26.6 | 103.6 | 30.8 | 104.3 | 27.8 |
| Medication (n/%) | | | | | | | | | | | | |
| Insulin | 0 | 0 | 1,521 | 3.6 | 1,521 | 1.6 | 0 | 0 | 47 | 3.9 | 47 | 1.0 |
| Non-insulin hypoglycemic agents | 0 | 0 | 7,752 | 18.2 | 7,752 | 7.9 | 0 | 0 | 255 | 21.1 | 255 | 5.6 |
| Antihypertensives | 39,605 | 71.8 | 14,948 | 35.2 | 54,553 | 55.9 | 225 | 6.7 | 112 | 9.3 | 337 | 7.4 |
| Statins | 10,967 | 19.9 | 3,215 | 7.6 | 14,182 | 14.5 | 141 | 4.2 | 127 | 10.5 | 268 | 5.9 |
| Medical history (n/%) | | | | | | | | | | | | |
| Atrial fibrillation | 2,339 | 4.2 | 705 | 1.7 | 3,044 | 3.1 | - | - | - | - | - | - |
| Myocardial infarction | 1,856 | 3.4 | 654 | 1.5 | 2,510 | 2.6 | - | - | - | - | - | - |
| Ischemic heart disease | 3,602 | 6.5 | 1,178 | 2.8 | 4,780 | 4.9 | 396 | 11.8 | 189 | 15.7 | 585 | 12.8 |
| Heart failure | 1,461 | 2.6 | 698 | 1.6 | 2,159 | 2.2 | - | - | - | - | - | - |
| Cerebrovascular disease | 5,056 | 9.2 | 1,693 | 4 | 6,749 | 6.9 | 63 | 1.9 | 41 | 3.4 | 104 | 2.3 |
| Peripheral vascular disease | 363 | 0.7 | 181 | 0.4 | 544 | 0.6 | - | - | - | - | - | - |
| Neuropathy | 71 | 0.1 | 91 | 0.2 | 162 | 0.2 | - | - | - | - | - | - |
| Amputation | 25 | 0 | 67 | 0.2 | 92 | 0.1 | - | - | - | - | - | - |
| Renal failure | 467 | 0.8 | 310 | 0.7 | 777 | 0.8 | 177 | 5.3 | 88 | 7.3 | 265 | 5.8 |
| Retinopathy | 693 | 1.3 | 345 | 0.8 | 1,038 | 1.1 | - | - | - | - | - | - |
| Cataract | 4,542 | 8.2 | 1,838 | 4.3 | 6,380 | 6.5 | 62 | 1.8 | 44 | 3.6 | 106 | 2.3 |
| Ulcer of skin | 201 | 0.4 | 161 | 0.4 | 362 | 0.4 | - | - | - | - | - | - |

BMI, body mass index; CHARLS, China Health and Retirement Longitudinal Study; CMS, Clinical Management System; eGFR, estimated glomerular filtration rate; HbA1c, glycosylated hemoglobin type A1c; HDL, high-density lipoprotein; LDL, low-density lipoprotein; SD, standard deviation.

for discrimination ranged from 0.636 (for retinopathy) to 0.813 (for amputation and renal failure). Calibration slopes between expected and observed outcome rates ranged from 0.680 to 1.333 for mortality, myocardial infarction, ischemic heart disease, retinopathy, neuropathy, ulcer of the skin, cataract, renal failure, and heart failure; 0.591 for peripheral vascular disease; 1.599 for cerebrovascular disease; and 2.247 for amputation (ideal = 1). All calibration intercepts ranged from −0.066 to 0.022 (ideal = 0; Table 2, Fig 2). The performance of the risk

**Table 2. Internal and external validation statistics for CHIME, UKPDS-OM2, and RECODe for diabetes.**

| | Internal validation: CMS[a] | | External validation: CHARLS | | UKPDS-OM2 | | RECODe | |
|---|---|---|---|---|---|---|---|---|
| | Discrimination: C-statistic (95% CI) | Calibration: slope/intercept | Discrimination: C-statistic (95% CI) | Calibration: slope/intercept | Discrimination: C-statistic (95% CI) | Calibration: slope/intercept | Discrimination: C-statistic (95% CI) | Calibration: slope/intercept |
| Mortality | 0.782 (0.769, 0.790) | 1.080/0.008 | 0.748 (0.692, 0.797) | 1.035/−0.009 | 0.750 (0.738, 0.757) in CMS, 0.750 (0.707, 0.798) in CHARLS | 0.777/−0.041 in CMS, 0.478/0.018 in CHARLS | 0.751 (0.740, 0.759) in CMS | 1.514/0.055 in CMS |
| Myocardial infarction | 0.770 (0.746, 0.790) | 0.827/−0.006 | - | - | 0.672 (0.650, 0.691) in CMS | 0.295/0.021 in CMS | 0.729 (0.711, 0.750) in CMS | 0.439/0.015 in CMS |
| Ischemic heart disease | 0.697 (0.684, 0.709) | 1.002/−0.022 | 0.734 (0.700, 0.768) | 0.874/0.082 | 0.572 (0.555, 0.594) in CMS, 0.517 (0.442, 0.534) in CHARLS | 0.669/0.072 in CMS, −0.589/0.191 in CHARLS | | |
| Heart failure | 0.802 (0.787, 0.821) | 0.789/0.005 | - | - | 0.711 (0.695, 0.732) in CMS | 0.780/−0.011 in CMS | 0.793 (0.777, 0.809) in CMS | 0.770/0.041 in CMS |
| Cerebrovascular disease | 0.722 (0.705, 0.736) | 1.599/−0.059 | 0.748 (0.616, 0.929) | -[b] | 0.666 (0.650, 0.684) in CMS, 0.514 (0.373, 0.616) in CHARLS | 0.359/0.053 in CMS, -[b] | 0.664 (0.645, 0.680) in CMS | 0.597/0.104 in CMS |
| Peripheral vascular disease | 0.770 (0.741, 0.801) | 0.591/−0.004 | - | - | | | | |
| Neuropathy | 0.744 (0.703, 0.788) | 1.113/−0.001 | - | - | | | 0.710 (0.671, 0.742) in CMS | 0.138/0.004 in CMS |
| Amputation | 0.813 (0.764, 0.861) | 2.247/−0.004 | - | - | 0.665 (0.597, 0.725) in CMS | 0.130/0.002 in CMS | | |
| Ulcer of skin | 0.763 (0.741, 0.789) | 1.124/0.006 | - | - | 0.548 (0.516, 0.578) in CMS | 0.260/0.041 in CMS | | |
| Renal failure | 0.813 (0.800, 0.826) | 0.680/−0.009 | 0.702 (0.652, 0.746) | 0.898/0.088 | 0.772 (0.752, 0.789) in CMS, 0.520 (0.477, 0.549) in CHARLS | 3.846/0.057 in CMS, 11.411/0.099 in CHARLS | 0.737 (0.719, 0.757) in CMS | 1.895/−0.053 in CMS |
| Cataract | 0.716 (0.703, 0.728) | 1.333/−0.066 | 0.770 (0.711, 0.854) | 0.709/−0.022 | | | | |
| Retinopathy | 0.636 (0.611, 0.659) | 0.971/0.020 | - | - | | | 0.615 (0.599, 0.638) in CMS | 0.290/0.057 in CMS |
| **RMSPE/R²** | | 3.53/0.934 | | 4.49/0.816 | | 10.82/0.532 in CMS, 14.80/0.227 in CHARLS | | 11.16/0.001 in CMS |

[a]CMS participants enrolled from 2006–2008.

[b]Only 3 centiles for calibration.

CHARLS, China Health and Retirement Longitudinal Study; CHIME, Chinese Hong Kong Integrated Modeling and Evaluation; CI, confidence interval; CMS, Clinical Management System; RECODe, Risk Equations for Complications Of type 2 Diabetes; RMSPE, root mean square percentage error; UKPDS-OM2, UK Prospective Diabetes Study Outcomes Model 2.

prediction models from internal validation overall for prediabetes and diabetes and their functional form are presented in S5 Table. Model performance did not vary substantially when evaluating participants with diabetes compared to prediabetes (S8 Table; S1 Fig) except for renal failure among prediabetes. In external validation against CHARLS data, CHIME predictions had C-statistics of 0.748 for mortality, 0.734 for ischemic heart disease, 0.748 for cerebrovascular disease, 0.702 for renal failure, and 0.770 for cataract. Calibration slopes ranged from 0.709 to 1.035 and intercepts between −0.022 and 0.088.

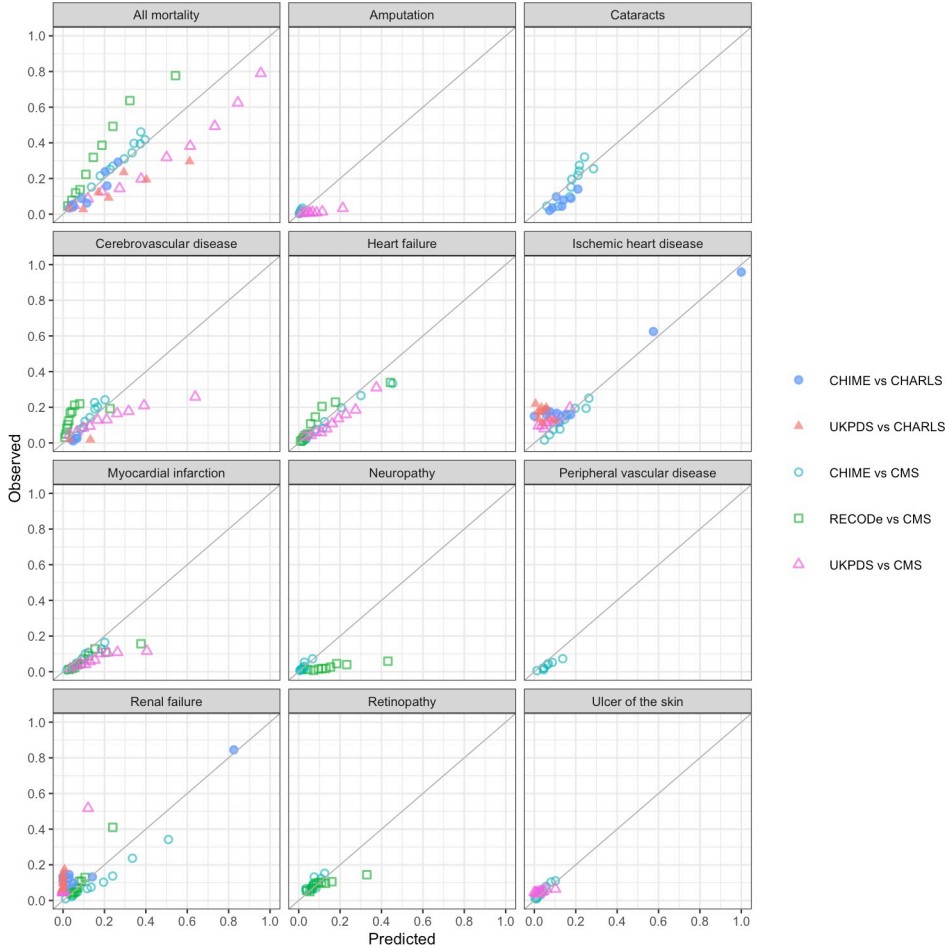

**Fig 2. Calibration plots of CHIME, UKPDS-OM2, and RECODe model for diabetes.** Predictions using CHIME, UKPDS-OM2, and RECODe are presented if available. Points are displayed for deciles of predicted and observed event rates, with fewer centiles than deciles used if fewer than 5 events were observed per group to prevent unstable inferences. CHARLS, China Health and Retirement Longitudinal Study; CHIME, Chinese Hong Kong Integrated Modeling and Evaluation; CMS, Clinical Management System; RECODe, Risk Equations for Complications Of type 2 Diabetes; UKPDS-OM2, United Kingdom Prospective Diabetes Study Outcomes Model 2.

## Comparison with alternative risk equations

Our CHIME model had better discrimination and calibration than UKPDS-OM2 in both the CMS development cohort (C-statistics 0.548 to 0.772, slopes 0.130 to 3.846, and intercepts −0.041 to 0.072) and CHARLS validation cohort (C-statistics 0.514 to 0.750, slopes −0.589 to 11.411, and intercepts 0.018 to 0.191; Table 2; Fig 2). CHIME had small improvements in discrimination and better calibration than RECODe for all outcomes in the CMS development cohort (C-statistics 0.615 to 0.793, slopes 0.138 to 1.514, and intercepts −0.053 to 0.104). The predictive error was smaller for CHIME against the CMS development data (RSMPE 3.53% versus 10.82% for UKPDS-OM2 and 11.16% for RECODe at 10 years of follow-up; Table 2), and the CHARLS validation cohort (RSMPE 4.49% versus 14.80% for UKPDS-OM2 at 6 years of follow-up).

The predicted event rates from UKPDS and CHIME models against the observed events rates over time for CMS participants with diabetes and prediabetes are shown in Figs 3 and S2 (RECODe model only has risk estimates at 10 years). On external validation, the UKPDS-OM2

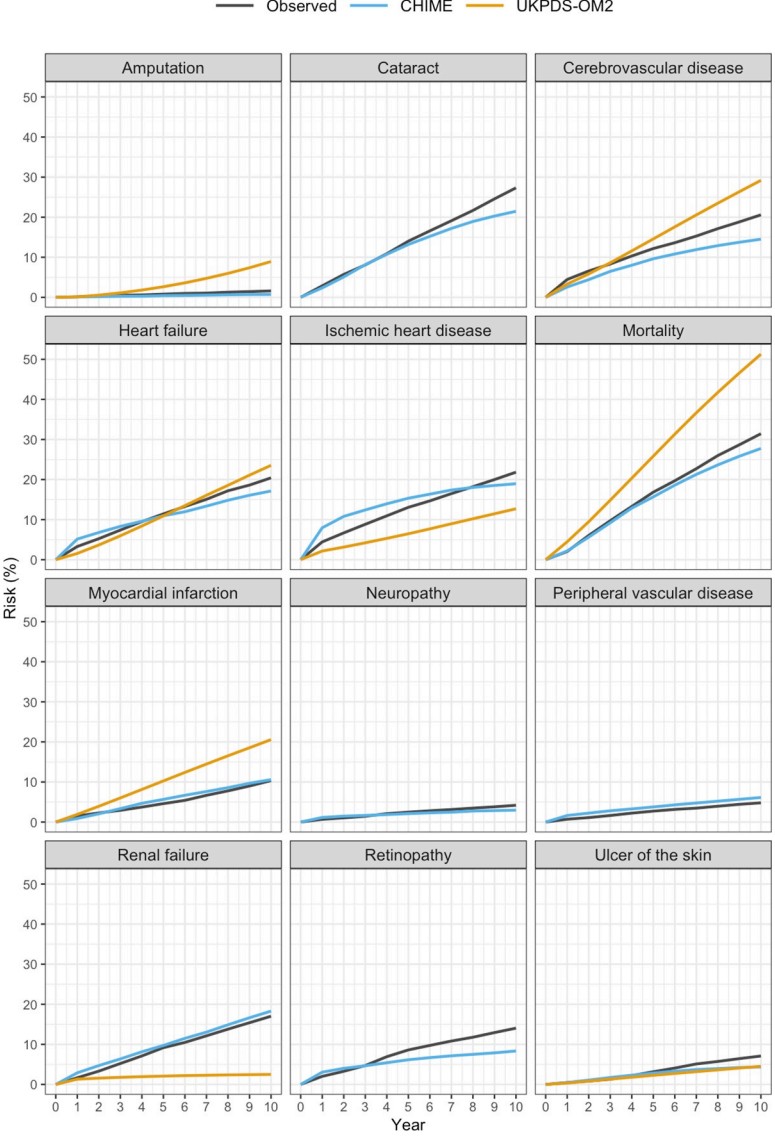

**Fig 3. Validation of CHIME and UKPDS-OM2 models against CMS cohort.** Predicted event rates from UKPDS and CHIME models against the observed events rates over time for CMS participants with diabetes. CHIME, Chinese Hong Kong Integrated Modeling and Evaluation; CMS, Clinical Management System; UKPDS-OM2, UK Prospective Diabetes Study Outcomes Model 2.

overpredicted 4 outcomes (mortality, myocardial infarction, cerebrovascular disease, and amputation), underpredicted 2 outcomes (ischemic heart disease and renal failure), and had close correspondence with heart failure and ulcer of the skin. The CHIME model was derived from the CMS data and displayed close correspondence on internal validation with the exception of development of diabetes.

Table 3 shows the validation of the CHIME model against 80 observed endpoints from 9 published trials. All simulation trial cohorts were checked for convergence of outcomes (S3 Fig). Among the simulated trial cohorts, the calibration performance of the CHIME model was generally better for trials with mainly Asian participants (RMSPE 0.48% to 3.66%, ideal = 0%) than for non-Asian trials (RMPSE 0.81% to 8.50%), with the exception of ADVANCE (RMSPE 0.81%) among non-Asian trials, although ADVANCE had a significant

**Table 3. Validation of CHIME, UKPDS-OM2, and RECODE models against simulated cohorts from trials.**

| Trial | Follow-up, year | Location/Ethnicity | Participants | Outcomes | CHIME | | UKPDS | | RECODe | |
|---|---|---|---|---|---|---|---|---|---|---|
| **Diabetes** | | | | | RMSPE (%) | $R^2$ | RMSPE (%) | $R^2$ | RMSPE (%) | $R^2$ |
| **JPAD, 2017** [28] (*n* = 2,539) | 10.3 | Japanese | Age 30–85 with type 2 diabetes and without preexisting CVD, recruited from 2002–2005 | Ischemic heart disease, myocardial infarction, cerebrovascular disease | 0.48 | 0.992 | 3.75 | 0.671 | 5.36 | 0.957 |
| **J-EDIT, 2012** [27] (*n* = 1,173) | 6 | Japanese | Age 65–85 with type 2 diabetes, recruited from 2001–2002 | Heart failure, ischemic heart disease, myocardial infarction, cerebrovascular disease, ulcer of the skin | 3.66 | 0.09 | 3.08 | 0.247 | 10.30 | 0.566 |
| **JDCS, 2010** [26] (*n* = 2,033) | 7.8 | Japanese | Age 40–70 with type 2 diabetes, recruited from 1995–1996 | Mortality, ischemic heart disease, myocardial infarction, cerebrovascular disease | 1.66 | 0.631 | 3.35 | 0.017 | 3.39 | 0.043 |
| **ACCORD, 2010** [23] (*n* = 4,733) | 4.7 | United States, Canada; non-Hispanic white (61%), Black (24%), Hispanic (7%) | Age 40–79 with type 2 diabetes and CVD, or ages 55–79 with substantial atherosclerosis, albuminuria, left ventricular hypertrophy, or at least 2 CVD risk factors, recruited in 2001, and 2003–2005 | Mortality, heart failure, ischemic heart disease, myocardial infarction, renal failure, cerebrovascular disease | 3.71 | 0.06 | 4.94 | 0.001 | 3.08 | 0.953 |
| **ADVANCE, 2007** [24] (*n* = 11,140) | 4.3 | Europe (46%), Asia (37%), Australia and New Zealand (13%), North America (4%); Asian (38.1%), White European (60.0%), Other (1.9%) | Age ≥55 with type 2 diabetes, and with a history of, or risk factor for CVD, from 20 countries, recruited from 2001–2003 | Mortality, ischemic heart disease, retinopathy, cerebrovascular disease | 0.81 | 0.849 | 4.15 | 0.066 | 6.10 | 0.772 |
| **UKPDS 33, 1998** [30] (*n* = 3,867) | 10 | United Kingdom; White (81%), Indian Asian (10%), Afro-Caribbean (8%), Other (1%) | Age 25–65 newly diagnosed patients with type 2 diabetes recruited from 1977–1991 | Mortality, amputation, cataracts, heart failure, myocardial infarction, renal failure, retinopathy, cerebrovascular disease | 6.62 | 0.127 | 5.09 | 0.986 | 7.19 | 0.111 |
| **UKPDS 80, 2008** [29] (*n* = 4,209) | 10 | United Kingdom; White (81%), Asian Indian (10%), Afro-Caribbean (9%) | Age 25–65 with newly diagnosed type 2 diabetes recruited from 1977–1991 | Mortality, myocardial infarction, peripheral vascular disease, cerebrovascular disease | 8.50 | 0.841 | 4.54 | 0.986 | 11.10 | 0.673 |
| **Prediabetes** | | | | | | | | | | |
| **ACE, 2017** [22] (*n* = 6,522) | 5 | Chinese | Prediabetes Age ≥50 with established CHD and impaired glucose tolerance, recruited from 2009–2015 | Mortality, diabetes status, ischemic heart disease, myocardial infarction, renal failure, cerebrovascular disease | 1.07 | 0.969 | - | - | - | - |
| **DPP, 2002** [25] (*n* = 3,234) | 6 | United States; White (55%), African American (20%), Hispanic (16%), American Indian (5%), Asian (4%) | Age ≥25 with BMI of ≥24 and a plasma fasting glucose of 5.3 to 6.9 mmol/L, and without diabetes, recruited from 1996–1999 | Development of diabetes | 3.19 | 0.940 | - | - | - | - |

*Simulated 100,000 participants for validation.

RMSPE, root mean square percentage error; CHD, coronary heart disease; CVD, cardiovascular disease; BMI, body mass index; eGFR, estimated glomerular filtration rate; ACE, Acarbose Cardiovascular Evaluation [22]; ACCORD, Action to Control Cardiovascular Risk in Diabetes [23]; ADVANCE, Action in Diabetes and Vascular disease: preterAx and diamicroN-MR Controlled Evaluation [24]; DPP, Diabetes Prevention Program [25]; JDCS, Japan Diabetes Complications Study [26]; J-EDIT, Japan Elderly Diabetes Intervention Trial [27]; JPAD, Japanese Primary Prevention of Atherosclerosis with Aspirin for Diabetes trial [28]; UKPDS, United Kingdom Prospective Diabetes Study [29,30].

number of Asian participants (Table 3). Compared to UKPDS-OM2 and RECODe, CHIME was the best performing model for JPAD, JDCS, ADVANCE, and comparable to UKPD-S-OM2 for J-EDIT (RMPSE 3.66% versus 3.08%). The best performing model for UKPDS and ACCORD trials were the respective models that used this for development—UKPDS-OM2 and RECODe. CHIME was the closest model for Asian trials (RMSPE 1.86%), and UKPD-S-OM2 was closest for European and North American trials (4.72%). Calibration performance of each model by clinical outcome is shown in S9 Table and Fig 4, and by individual outcome for each individual trial in S10 Table.

## Output

We developed an online, public, interactive interface for modeling diabetes and prediabetes outcomes, allowing input of demographic and clinical information to calculate risk

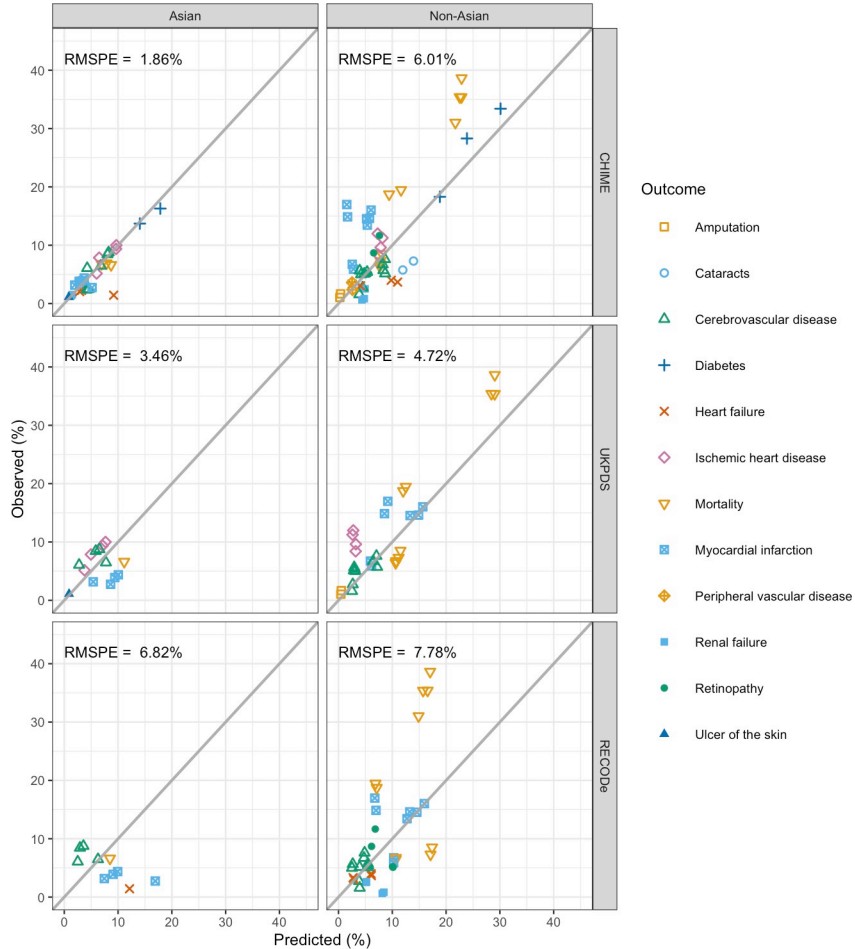

**Fig 4. External validation of CHIME, UKPDS-OM2, and RECODe against trial data.** Predicted percentage of events from CHIME, UKPDS-OM2, and RECODe against observed percentage in Asian trials (ACE, JPAD, J-EDIT, and JDCS) and non-Asian trials (ACCORD, ADVANCE, DPP, UKPDS 33, and UKPDS 80). See Table 3 for further details of trials. ACCORD, Action to Control Cardiovascular Risk in Diabetes; ACE, Acarbose Cardiovascular Evaluation; ADVANCE, Action in Diabetes and Vascular disease: preterAx and diamicroN-MR Controlled Evaluation; CHIME, Chinese Hong Kong Integrated Modeling and Evaluation; DPP, Diabetes Prevention Program; JDCS, Japan Diabetes Complications Study; J-EDIT, Japan Elderly Diabetes Intervention Trial; JPAD, Japanese Primary Prevention of Atherosclerosis with Aspirin for Diabetes; RECODe, Risk Equations for Complications Of type 2 Diabetes; RMSPE, root mean square percentage error; UKPDS-OM2, UK Prospective Diabetes Study Outcomes Model 2; UKPDS 33, UK Prospective Diabetes Study 33; UKPDS 80, UK Prospective Diabetes Study 80.

probabilities. Full risk equation formulas and data visualization are presented online: https://jquan.shinyapps.io/CHIME.

## Discussion

In the current study, we developed and externally validated the first integrated prediabetes and type 2 diabetes outcomes model for Chinese and East Asian populations: comprising of 13 outcomes including mortality, micro- and macrovascular complications, and development of diabetes. We validated using both individual-level data in the CHARLS cohort and aggregate-level data using simulated cohorts from 9 published trials. We compared the CHIME model to the existing UKPDS-OM2 and RECODe models.

We found that the widely used UKPDS-OM2 was not well calibrated to the Chinese population on external validation of 2 individual-level datasets. The UKPDS-OM2 was developed from a 1970s UK cohort and overpredicted mortality and cerebrovascular disease but underpredicted outcomes that are more common in the Asian population such as renal failure, reflecting the differences in epidemiology of diabetes between East Asian and European/North American populations [3]. The RECODe model developed from a North American trial displayed similar patterns of overpredicting myocardial infarction and cerebrovascular disease but underpredicting renal failure. The overprediction of macrovascular outcome by UKPDS-OM2 could be due to more intensive management in the past decade, such as early initiation and tighter clinical thresholds for antihyperglycemic agents, statins and antihypertensives, whereas RECODe was developed from a more recent trial conducted from 2001 to 2009 and consequently better calibration than UKPDS-OM2.

The RECODe model showed good discrimination, often comparable to the CHIME model, but was less well calibrated to the CMS development cohort. Unlike CHIME and UKPDS-OM2, the RECODe model is restricted to predicting risk at a specific time interval of 10 years and does not incorporate time-varying covariates. This limits its applicability for lifetime projections, and flexibility when validating against sample of varying follow-up periods. As expected, the best performing model for cohorts simulated from UKPDS trial was UKPDS-OM2, and for ACCORD trial was RECODe, which were their respective development cohorts, supporting the face validity of this validation approach. The RECODe (ACCORD trial) differs markedly from the social and historical context of UKPDS trial [7], both of which differ even more markedly from the Chinese CMS and CHARLS cohorts.

The CHIME model showed good calibration between predicted and observed probabilities by deciles of risk against the CMS development cohort for most outcomes (mortality, myocardial infarction, ischemic heart disease, ulcer of the skin, retinopathy, neuropathy, renal failure, and heart failure) with poorer calibration among the higher-risk subgroups for cerebrovascular disease, peripheral vascular disease, and amputation. The CHIME model had good calibration against CHARLS, whereas the UKPDS OM-2 was poorly calibrated.

While in general CHIME performed better in Asian than non-Asian trials, there were 2 notable exceptions in J-EDIT and ADVANCE. The J-EDIT trial enrolled substantially older participants of age 65 to 84 years (mean age 72) compared to the CMS dataset (mean age 61.9). This older age group with higher risk may be closer to the UKPDS-OM2, which tends to overestimate risks in trials and cohorts [4,6,7,24]. CHIME also performed well for the ADVANCE trial, which may be due to the diverse geographical/ethnic mix of the trial participants with almost 40% of participants from Asia.

Among the various outcomes, the prediction of diabetes status and impaired renal function for participants with prediabetes was notably worse, likely due to the insidious onset of diabetes and impaired renal function and the lack of routine screening at regular intervals in

population-based cohorts. More accurate ascertainment was achieved in prediabetes trial settings such as ACE, which employ more rigorous ascertainment of the development of diabetes and impaired renal function as an outcome. Overall calibration was good for the trials supporting population-level policy assessment and health economic evaluation. The CHIME models can be useful for population-level risk prediction of several endpoints considered together rather than at the individual level.

Previous diabetes simulation models are typically developed from trial data on limited number of participants in European or North American settings and were not externally validated on an individual-level dataset. Study participants in trials are selected according to strict inclusion and exclusion criteria that may not reflect real-world generalizability that is essential for useful policy modeling of whole populations. In contrast, we used an extensive population-based health records database based on routine health contact that has the benefit of a larger sample size and generalizability. We conducted validation against real-world observational data from a nationally representative CHARLS cohort. Similar to other validation studies of diabetes outcomes models, we also generated simulated cohorts from reported data on trial participants to further validate against additional outcomes [8–10]. Due to varying inclusion and exclusion criteria and trial protocol-driven practices among different studies, there are likely to be differences between the simulated cohort generated and the characteristics of the actual patients in the trial.

The CHIME model was developed according to American Diabetes Association and ISPOR guidelines on modeling best practice [40–42]. Similar to other diabetes modeling approaches to uncertainty and heterogeneity, we addressed first-order uncertainty (stochastic uncertainty) by performing Monte Carlo simulations with sufficient replications for convergence, second-order uncertainty (parameter uncertainty) by bootstrap resampling with replacement of individuals in the study population and reestimating equations to derive a distribution of parameters for each equation, and patient heterogeneity by using individual-level simulation of a large sample [10,43].

Our study had a number of limitations. While we modeled a broad range of diabetes-related outcomes, some complications such as hypoglycemic episodes were not able to be included due to lack of data. The development sample was drawn from population-based health records, which have less complete ascertainment of clinical data compared to the idealized settings of clinical trials. Nevertheless, our sample size was far larger—almost 10-fold higher than the UKPDS ($n$ = 5,102) and ACCORD trials ($n$ = 10,251) utilized to develop previous diabetes prediction models [5,6,8–10]. Our electronic medical records covered all public healthcare services across the territory, which increases generalizability compared to the strict inclusion and exclusion criteria of the randomized trials used in the development of other diabetes models [5,8–10]. There was a general lack of long-term Chinese- or East Asian–specific cohorts or trials longer than 4 years. We excluded the China Da Qing Diabetes Prevention Study (CDQDPS) study of a 1980s Chinese cohort due its small sample size and lack of baseline biomarkers [44] and the Japan Diabetes Optimal Treatment study for 3 major risk factors of cardiovascular diseases (J-DOIT3) trial as it only published composite endpoints [45]. The neuropathy endpoint was unavailable from the trial data. Some trials failed to report sufficient details such as rates of existing complications at baseline. Further work on validating against more outcomes, longer follow-up, and in other East Asian populations are warranted. New predictive biomarkers such as hs-CRP and serum amyloid P component have improved for mortality prediction for diabetes, and their inclusion may improve predictions for other outcomes [46,47].

In many health systems, access to interventions is often dependent on evidence of value for money. For diabetes, this will require simulation modeling to estimate the long-term health outcomes and to inform decision analysis such as cost per quality-adjusted life years (QALYs)

gained. Estimation of inputs including complication-related costs, healthcare utilization, and health state utility values will require further work for East Asia settings. The CHIME outcomes model can be used to evaluate population health status for prediabetes and diabetes using routinely recorded data. By applying the appropriate utility values of the target population for the wide range of diabetes-related complications [48], the CHIME outcomes model can be utilized to assess quality of life and measure QALYs over the long-time horizon of chronic disease conditions. This supports economic evaluation of policy guidelines and clinical treatment pathways to tackle diabetes, prediabetes, their associated micro- and macrovascular complications, and life expectancy.

Our study shows that the CHIME model is a validated tool for predicting progression of diabetes and its outcomes, particularly among Chinese and East Asian populations, which has been lacking thus far. This will support the clinical and economic evaluation of therapies related to the long-term management of diabetes. The CHIME model can be used by health service planners and policy makers to develop population-level strategies, for example, setting HbA1c and lipid targets, to optimize health outcomes.

## Supporting information

**S1 TRIPOD Checklist.**
(DOCX)

**S1 Fig. Calibration plots in prediabetes population.** Predictions using CHIME are presented if available. Points are displayed for deciles of predicted and observed event rates, with fewer centiles than deciles used if fewer than 5 events were observed per group to prevent unstable inferences. CHARLS, China Health and Retirement Longitudinal Study; CHIME, Chinese Hong Kong Integrated Modeling and Evaluation; CMS, Clinical Management System.
(DOCX)

**S2 Fig. Validation of CHIME against CMS cohort with prediabetes.** Predicted event rates from CHIME model against the observed events rates over time for CMS participants with prediabetes. CHIME, Chinese Hong Kong Integrated Modeling and Evaluation; CMS, Clinical Management System.
(DOCX)

**S3 Fig. Convergence plot of trial simulations.** See Table 3 for further details of trials. ACCORD, Action to Control Cardiovascular Risk in Diabetes; ACE, Acarbose Cardiovascular Evaluation; ADVANCE, Action in Diabetes and Vascular disease: preterAx and diamicroN-MR Controlled Evaluation; DPP, Diabetes Prevention Program; JDCS, Japan Diabetes Complications Study; J-EDIT, Japan Elderly Diabetes Intervention Trial; JPAD, Japanese Primary Prevention of Atherosclerosis with Aspirin for Diabetes; UKPDS 33, UK Prospective Diabetes Study 33; UKPDS 80, UK Prospective Diabetes Study 80.
(PDF)

**S1 Table. Diabetes outcomes prediction models.**
(DOCX)

**S2 Table. Definition of outcomes used in the model by International Classification of Disease (ICD-9) and International Classification of Primary Care (ICPC-2) coding.**
(DOCX)

**S3 Table. Missing data at baseline for CMS dataset.**
(DOCX)

**S4 Table. Missing data at follow-up for CHARLS validation cohort.**
(DOCX)

**S5 Table. Internal validation of CHIME prediction models on CMS cohort 2006 to 2017 (Diabetes and Prediabetes).**
(DOCX)

**S6 Table. Coefficients of predictors for CHIME biomarkers.**
(DOCX)

**S7 Table. Survival time ratios of predictors in the CHIME risk equations.**
(DOCX)

**S8 Table. Validation statistics for CHIME for participants with prediabetes.**
(DOCX)

**S9 Table. External validation of observed against predicted trial endpoints across all validation trials by outcome for CHIME, UKPDS-OM2, and RECODe models.**
(DOCX)

**S10 Table. External validation of observed against predicted endpoints by outcome and individual trial for CHIME, UKPDS-OM2, and RECODe models.**
(DOCX)

## Acknowledgments

The authors thank the Hospital Authority for kindly providing the CMS data. We thank the China Center for Economic Research and the National School of Development of Peking University for providing the CHARLS data.

J.Q. and C.S.N. are the guarantors of this work and, as such, had full access to all the data in the study and take responsibility for the integrity of the data and the accuracy of the data analysis.

## Author Contributions

**Conceptualization:** Jianchao Quan, Carmen S. Ng, Gabriel M. Leung.

**Data curation:** Jianchao Quan, Carmen S. Ng, Harley H. Y. Kwok, Ada Zhang, Yuet H. Yuen.

**Formal analysis:** Jianchao Quan, Carmen S. Ng, Karen Eggleston, Gabriel M. Leung.

**Funding acquisition:** Jianchao Quan.

**Investigation:** Jianchao Quan, Carmen S. Ng, Ada Zhang, Yuet H. Yuen.

**Methodology:** Jianchao Quan, Carmen S. Ng.

**Project administration:** Jianchao Quan, Harley H. Y. Kwok.

**Resources:** Jianchao Quan, Cheung-Hei Choi, Shing-Chung Siu, Simon Y. Tang, Nelson M. Wat, Jean Woo, Karen Eggleston, Gabriel M. Leung.

**Software:** Jianchao Quan, Carmen S. Ng.

**Supervision:** Jianchao Quan, Gabriel M. Leung.

**Validation:** Jianchao Quan, Carmen S. Ng.

**Visualization:** Jianchao Quan, Carmen S. Ng, Harley H. Y. Kwok.

**Writing – original draft:** Jianchao Quan, Carmen S. Ng.

**Writing – review & editing:** Jianchao Quan, Carmen S. Ng, Harley H. Y. Kwok, Ada Zhang, Yuet H. Yuen, Cheung-Hei Choi, Shing-Chung Siu, Simon Y. Tang, Nelson M. Wat, Jean Woo, Karen Eggleston, Gabriel M. Leung.

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
