## [Editor Report · Decision Letter 0]

7 Dec 2020

Dear Dr Ng, 

Thank you for submitting your manuscript entitled "Development and validation of a patient-level model to assess lifetime health outcomes of prediabetes and type 2 diabetes in Chinese populations (CHIME) a modelling study" for consideration by PLOS Medicine.

Your manuscript has now been evaluated by the PLOS Medicine editorial staff, as well as by an academic editor with relevant expertise, and I am writing to let you know that we would like to send your submission out for external peer review.

Kind regards,

Caitlin Moyer, Ph.D.,

Associate Editor

PLOS Medicine

---

## [Decision Letter · Decision Letter 1]

9 Feb 2021

Dear Dr. Ng,

Thank you very much for submitting your manuscript "Development and validation of a patient-level model to assess lifetime health outcomes of prediabetes and type 2 diabetes in Chinese populations (CHIME): a modelling study" (PMEDICINE-D-20-05652R1) for consideration at PLOS Medicine. 

Your paper was evaluated by a senior editor and discussed among all the editors here. It was also discussed with an academic editor with relevant expertise, and sent to three independent reviewers, including a statistical reviewer. The reviews are appended at the bottom of this email and any accompanying reviewer attachments can be seen via the link below:

[LINK]

In light of these reviews, I am afraid that we will not be able to accept the manuscript for publication in the journal in its current form, but we would like to consider a revised version that addresses the reviewers' and editors' comments. Obviously we cannot make any decision about publication until we have seen the revised manuscript and your response, and we plan to seek re-review by one or more of the reviewers. 

We expect to receive your revised manuscript by Mar 02 2021 11:59PM. Please email us (plosmedicine@plos.org) if you have any questions or concerns.

We look forward to receiving your revised manuscript. 

Sincerely,

Emma Veitch, PhD

PLOS Medicine

On behalf of Caitlin Moyer, PhD

Associate Editor, PLOS Medicine

plosmedicine.org

*Comments from reviewers are generally positive on the study, but reviewers did request quite a bit of clarification on points including: comparing the model to the RECODe model and ENFORCE mode, and UKPDS-OM2l, the fact that all 13 of the outcomes may not have been validated, reporting of associations between risk factors and clinical outcomes being counterintuitive in table S6; whether there should be separate prediction models for prediabetes and T2 diabetes, the issue of missing data at baseline.

*In your revised paper, please clarify if the analytical approach followed here corresponded to one laid out in a prospective protocol or analysis plan? Please state this (either way) early in the Methods section.

Comments from the reviewers:

Reviewer #1: "Development and validation of a patient-level model to assess lifetime health outcomes of prediabetes and type 2 diabetes in Chinese populations (CHIME): a modelling study" presents predictive outcome models for diabetic and prediabetic patients, developed from an East Asian (Hong Kong Chinese) population of nearly 100,000 participants. The developed Chinese Hong Kong Integrated Modelling and Evaluation (CHIME) model consists of 13 parametric proportional hazard model-based risk equations, each corresponding to one of thirteen outcomes (all-cause mortality, 4 macrovascular outcomes, 7 microvascular outcomes, development of diabetes). Calibration performance was performed on the development data and outperformed the UKPDS-OM2 significantly. External validation was pursued on the China Health and Retirement Longitudinal Study (CHARLS), and simulated data from 9 published trials (one Chinese, three Japanese, five Western). It was found that in general, CHIME performed better for the East Asian trial data, as compared to the Western data. Online interactive visualization and the risk equation formula were provided.

While the presentation is detailed and fairly comprehensive, there remain some concerns that might be clarified:

1. For the comparison of CHIME vs. UKPDS-OM2 (Table 2), the set of outcomes involved for each model are different (e.g. CHIME includes retinopathy, but not UKPDS-OM2), but a single (aggregated) RMSPE and R^2 are provided for each model. It might be clarified as to whether the RMSPE/R^2 metrics were computed only on the common set of outcomes shared between both models. Ideally, the metrics for each outcome for each model might be provided (as in Table S7)

2. For Table S7, trial endpoint metrics might also be provided for individual trials where appropriate, in addition to the aggregate values (for the multiple Japanese/Western trials), especially as it is not clear whether some outcomes might be available in some trials for an aggregate group.

3. While UKPDS-OM2 was compared against CHIME for the CMS derivation cohort (where CHIME's superior performance is perhaps expected) and an external CHARLS cohort, a more complete evaluation would also compare CHIME & UKPDS-OM2 on the simulated data from the 9 additional trials. This would better affirm whether CHIME's superior performance is restricted to non-Asian cohorts.

4. In Line 127, it is stated that individuals under the age of 20 at date of onset were excluded. It might be clarified as to whether the onset of diabetes and/or prediabetes is referenced here, and whether the exclusion by age is standard or has some other justification.

5. In Line 179, it is stated that the probability (for certain predictors) to be missing does not depend on outcome (for the CMS dataset used for model prediction). Whether this property of missing probability not depending on outcome was empirically confirmed from the CMS data might be explicitly stated, since the relevant citations appear to be to publications that support the use of complete case analysis assuming the stated property.

6. The model fitting process in the Statistical methods section (Lines 183-198) might be described in greater detail, possibly in the supplementary appendix. In particular, how did consultation from clinician experts potentially modify the models fitted by backwards selection? And how were models selected by AIC for multiple distributions (exponential, log-logistic, log-normal, Weibull)?

7. For the CHIME model in Figure 1 (Lines 200-222), it is stated that each individual patient is simulated up until death or time horizon. However, from what could be understood, each risk equation gives a probability Prob{T>=t} given patient features, i.e. would return a probability of each outcome in a given year (e.g. 10% chance of mortality for the next year). As such, is the CHIME model stochastic (i.e. simulates a patient's death/outcomes according to a randomly generated value, and thus might produce different outcomes each time it is executed for the same patient), or is some cutoff implemented, e.g. on the threshold of each outcome?

8. For Figure 2, only 8 of the 13 outcomes are displayed. Calibration for CHIME vs. Observed risk might be shown for the remaining outcomes, regardless of availability of UKPDS-OM2 data.

9. For the simulation against published trials by modelling (Lines 247-267), the modelling procedure is described as generating each value from its reported mean and standard deviation. However, it appears likely that at least some variables are dependant on each other (e.g. for "duration of diabetes", its distribution might be dependant on/correlated with patient age for example, with older patients tending to have longer durations). It might be clarified as to whether such considerations were considered. If not, it might be considered to incorporate such dependancies by estimating from existing individual data.

10. For the online model at https://jquan.shinyapps.io/CHIME/, the predictors appear to have binary values (e.g. Age is either 50 or 60 only). Is this just an interface implementation choice (i.e. the model works when age is say 55 too)?

Reviewer #2: Comments on "Development and validation of a patient-level model to assess lifetime health outcomes of prediabetes and type 2 diabetes in Chinese populations (CHIME): a modelling study". There are some issues that the authors should take care of:

1. The authors thought that existing predictive outcomes models for type 2 diabetes developed and validated in historical Western populations may not be applicable for East Asian populations, so there is an urgent need to develop a model for predicting progression of diabetes and related outcomes in Chinese and East Asian populations. What is the predictive performance of UKPDS-OM2 for diabetes progression and related health outcomes in previous studies? Does this CHIME model in this study add other predictive variables compared to the UKPDS-OM2? 

2. Figure 1 shows that the individual's updated risk factors and history of events are used to predict the occurrence of outcomes and changes in risk factors in the next annual cycle. Why did this study not consider the changes of predictors and changes in medication use during follow-up (regular medication use or not, etc.) for the construction of the model?

3. The authors included diagnosed with either prediabetes or type 2 diabetes. Why were lifetime health outcome prediction models not developed and validated separately for prediabetes and type 2 diabetes? For the endpoint of diabetes, whether only prediabetes data were analyzed?

4. Table S3 shows the sample size of CMS database was 1,542,103 while the complete cases database was 97,628. The missing data among CMS participants at baseline, there are many indictors with a higher rate of missing (BMI, HbA1c, blood pressure, etc.). The updated patient characteristics predictors included in this study were used to develop the model rather than the baseline characteristics.

5. The inclusion and exclusion criteria for cohort studies and RCT are different and RCT has more stringent inclusion and exclusion criteria. The authors used the published baseline characteristics of the trial participants to generate a simulated cohort. For continuous variables, they took the reported mean and standard deviation to randomly generate values; for binary and categorical variables, they took the percentage of participants with the particular status as the sampling probability. Please describe the relevant references for this method.

6. Figure 2 shows that the CHIME model overpredicted mortality in CHARLS data, the author thought that the lower observed mortality rate attributed to incomplete ascertainment from loss of follow-up as deaths. What is the missing rate of each endpoint indicator during the follow-up of this study?

7. Table S6 shows the hazard ratios of predictors on outcome indicators. In this model, hazard ratio of age for each outcome indicators were less than 1 implying that age is a protective factor? Are the results of the indicators included in this study consistent with the results of the previous studies? For eGFR as a categorical indicator, which was used as the reference? This study included demographic indicators as predictors, and the authors should consider the issue of covariance between the individual predictor variables.

Reviewer #3: The Authors developed and validated a 13-equation simulation model (CHIME, Chinese Hong Kong Integrated Modeling and Evaluation) from a population-based cohort of individuals with prediabetes or type 2 diabetes (97,628 participants maturing 397,617 person-years) which predicts the individual risk of mortality, micro and macrovascular complications and the development of diabetes. External validation of CHIME was pursued in the CHARLS (China Health and Retirement Longitudinal Study; years 2011-2015) and in 100,000 simulated individuals from nine published studies (five of which carried out in Western countries). The data clearly indicate that CHIME is a new validated prediction model for the risk of progression from pre-diabetes to overt diabetes and the risk of diabetes outcomes. The Authors claim that CHIME performs better among Chinese than Japanese and Western populations, thus filling a gap due to the modest applicability among Asians of Western population-derived forecasting models available so far. The authors like to suggest that models such as theirs have the potential to be implemented by policy makers to plan the organization of health services and to possibly invent new strategies.

The paper is well written and sufficiently concise; most of the conclusions are consistent with the results reported. Overall, studies like this are welcome as they provide a well-functioning and validated predictive models that are likely to soon become a "service" to the health care systems and help make a precision medicine-approach a reality.

I have some criticisms referring to the alleged specificity that the authors attribute to their model.

In details:

1. The model used as a reference was the UKPDS-OM2, which is not the best available among those derived from Western populations. In fact, it is a pity the Authors did not consider testing RECODe, a new model reported to perform much better and to work well in samples from both clinical trials and real-life setting (Basu S et al, Lancet Diabetes Endocrinol 2017; 5: 788 and Basu S et al, Diabetes Care 2018; 41:586). The Authors should, therefore, compare CHIME to RECODe. The use of RECODe will also play the important function to investigate if models derived from non-Asian populations cannot be used to predict clinical outcomes in Asian individuals with diabetes, as the Authors suggest both in the Abstract and the Introduction. As far as predicting all-cause mortality is concerned (by far the most important clinical outcome) and along the same line as above, the Authors might be interested to compare their model also to ENFORCE an established, validated and parsimonious model (Copetti M et al, J Clin Endocrinol Metab 2019, 104: 4900) which has also proven to be easily improved by adding new markers (Scarale MG et al, Diabetes Care 2020, 43:1). Does CHIME outperform ENFORCE in Chinese individuals? As for RECODe, answering this question will improve a lot the relevance of this study. 

2. The Authors should report classical measures of calibration and discrimination for each of the 13 risk equations both in the development and in the validation cohorts. RMSPE and R2 which summarized all outcomes cannot be considered exhaustive. E.g. calibration in-the-large, calibration slope are usually used to assess a model's calibration; while survival c-statistic and other are usually used to assess a model's accuracy. RMSPE and R2 measure and overall calibration for all endpoints but, potentially, CHIME could be better than UKPDS-OM2 (or RECODe or ENFORCE) for certain endpoints and worse for other endpoints.

3. While 13 outcomes were used in model development, only five (mortality, heart disease, cerebrovascular disease, renal failure and diabetes status) were validated in the second external cohort. If the Author are not able to validate all 13 outcomes, this should be clearly said in the Abstract, results and Conclusions. 

4. In the validation of CHIME against simulated trial cohorts, a wide range of results was obtained. In fact, RMPS ranges 0.48-3.66 and 0.81-8.56 across trials in Japanese and non-Asians patients, respectively. Two of the five trials from the latter subgroup had RMPS well within the range of values observed in Japanese. Do these heterogeneous results allow us to state, as the Authors did (see Abstract and Conclusions), that the use of CHIME is somehow limited to the Chinese and East Asian populations? Any speculation on how to explain such a great variability? Again, addressing this point, like the one above, will increase the overall relevance of the study, placing it in the broader context of the portability of prediction models across diverse genetic, environmental and cultural backgrounds.

5. We have serious concerns about Table S6. In detail, the direction of nearly all associations between clinical outcome and established risk factors appears to be the exact opposite of what was expected. For example, the HR of age or duration of disease for mortality or cardiovascular events is <1, thus suggesting that the two risk factors are protective. The same goes for kidney function. How is it possible? We can speculate that these seemingly counterintuitive results depend on the way HRs were calculated, but unfortunately nothing is said in the methods to help address this problem.

6. Validation of the CHIME model against additional results (eighty) was performed in nine simulated trial cohorts. This cannot be considered an appropriate validation step. Also, if 13 risk equations were developed, how were 80 endpoints (and thus 80 risk equations) validated? In all, this piece of data should be removed.

Minor

A. The two samples used for discovery (CMS) and validation (CHARLES) are quite different in terms of several characteristics known to model diabetes risk and its complications, including the rate of prediabetes (51 vs 74%, respectively), current smokers (11 vs 29%), anti-hypertensive treatment (56 vs 22%), statin treatment (14 vs 6%) and, finally, in terms of eGFR (89 vs 103 ml / min). Despite all this, CHIME performs equally well in the two samples, a result that clearly speaks in favor of its portability in different clinical settings. I strongly suggest the Authors to comment on this aspect. 

B. The time horizon used to estimate the survival C-statistic has to be reported in the Methods section.

C. C-statistics should be reported in Table 2 to help comparing different prediction models.

[LINK]

---

## [Decision Letter · Decision Letter 2]

5 Apr 2021

Dear Dr. Ng,

Thank you very much for submitting your manuscript "Development and validation of a patient-level model to assess lifetime health outcomes of prediabetes and type 2 diabetes in Chinese populations (CHIME): a modelling study" (PMEDICINE-D-20-05652R2) for consideration at PLOS Medicine. 

Your revised paper was evaluated by a senior editor and discussed among all the editors here. It was also discussed with an academic editor with relevant expertise, and sent to three of the original reviewers, including a statistical reviewer. The reviews are appended at the bottom of this email and any accompanying reviewer attachments can be seen via the link below:

[LINK]

In light of the remaining concerns pointed out by the reviewers, I am afraid that we will not be able to accept the manuscript for publication in the journal in its current form, but we would like to consider a second revised version that addresses the reviewers' and editors' comments. Obviously we cannot make any decision about publication until we have seen the revised manuscript and your response, and we plan to seek re-review by one or more of the reviewers. 

We expect to receive your revised manuscript by . Please email us (plosmedicine@plos.org) if you have any questions or concerns.

We look forward to receiving your revised manuscript. 

Sincerely,

Caitlin Moyer, PhD 

Associate Editor 

PLOS Medicine

plosmedicine.org

1.Author summary, Introduction, and throughout the text: Please avoid use of the term “Western” populations- please use European or another term that best fits your intended meaning.

2. Methods Lines 114-117: Please be more specific in mentioning these analyses in the published trials (were the 9 trials determined in advance, for example) and the further analyses and comparisons carried out in response to peer review if possible. “Initially we planned to validate against simulated cohorts from published trials before we obtained the individual level CHARLS data. Further analyses on model calibration and comparison with other models were made in response to peer review.”

3. Methods: Line 119: Please fully define the IRB abbreviations here. 

4. Methods: Line 203-206: Please do not use underline for emphasis.

5. Table 1: Please include a descriptive legend, defining all abbreviations used in the table.

6. TRIPOD Checklist: Thank you for including the TRIPOD checklist. Please replace the page numbers in the checklist with section/paragraph numbers to refer to locations within the text.

7. Table 3: Please replace the term “Caucasian” with “White” in this table.

8. Table S9: Please define all abbreviations in the legend (IHD, MI, PVD).

Comments from the reviewers:

Reviewer #1: We thank the authors for addressing the points raised in the previous review round. Some additional clarifications might be considered:

1. Details on the Monte Carlo simulation convergence procedure for patient death simulation might be provided, perhaps in supplementary material.

2. The individual RMSPEs for each outcome/endpoint corresponding to each individual dataset (as apparently presented in Figure 4, and summarized in aggregate in S9 Table) might be provided, possibly in supplementary material.

Figure 4 might be reworked as there are currently multiple datapoints mapped to multiple outcomes indicated by approximately the same colour (e.g. Amputation/Mortality/Peripheral vascular disease); it might be advised to more clearly specify what each individual datapoint corresponds to (perhaps with differently-shaped datapoints, other markers)

3. In general, the eighty individual endpoint results from the CHIME, UKPDS-OM2 & RECODE validations (as aggregated in Table 3) might be reported, again possibly in supplementary material. While the aggregated RMSPEs provide a general idea of the model accuracy, it would probably be appropriate to know their applicability to individual outcomes.

4. The significant RMSPE differences for CHIME modelling, between the diabetes and prediabetes cohort (S9 Table), might be discussed. For example, for the mortality outcome, diabetes has RMSPE of 8.66 (inferior to UKPDS-OM2), compared to only 0.69 for prediabetes. Does this suggest that the CHIME model is particularly suited to prediabetes cases, and if so, might a separate model specialized towards diabetes cases be warranted?

Reviewer #2: Comments on "Development and validation of a patient-level model to assess lifetime health outcomes of prediabetes and type 2 diabetes in Chinese populations (CHIME): a modelling study". 

This study developed an individual-level health outcomes model (CHIME) and compares it with UKPDS-OM2 and RECODe models. CHIME model is a new validated tool for predicting progression of diabetes and its outcomes, particularly among Chinese and East Asian populations. I am satisfied with the author's revisions and recommend the manuscript for publication in the journal. But I still have a simple suggestion:

1. Table S7 explores the survival time ratios of predictors in CHIME risk equation, please provide a description of the statistical method used here. 

Reviewer #3: Major

1. The Authors have addressed some of my criticisms but not all of them. In details, calibration and discrimination measures were not reported when prediction models were applied in the simulated cohorts. This is mandatory given that, once again, it is not fair to propose prediction tools without providing their most intrinsic features. 

2. Unfortunately, some of the new findings question the quality of the proposed prediction tool. In fact, the calibration intercept is very far from being 0 (i.e. the optimal value) in both development and validation samples for many endpoints. Furthermore, the calibration slope is very far from 1 (i.e. the optimal value) in the validation samples for all endpoints. Overall, this clearly indicates that the prediction tool is not calibrated. Despite this frustrating result, the Authors have not adopted any strategy to correct it, nor have they commented on it fairly. Conversely, in the Discussion Section (lines 415-416) it is stated that "The CHIME model showed good calibration against the CMS development cohort, moderate calibration against CHARLS and better UKPDS OM-2 calibration." Again, this is simply not the case, as the calibration is not good in CMS and very poor in CHARLS. This is a serious flaw and must be discussed openly.

Minor

1. Surprisingly, the C-statistic for ischemic heart disease went from 0.697 in the development sample to 0.837 in the validation sample. A similar result was also found for the cerebrovascular outcome. This is an unexpected result, albeit a positive one, as the C-statistic is usually higher in the development sample. Authors should address and comment on this result.

2. When using UKPDS-OM2 and RECODe a good calibration is not shown for all results. This problem, which is especially surprising for the RECODe model, should also be adequately addressed and discussed.

[LINK]

---

## [Decision Letter · Decision Letter 3]

4 Jun 2021

Dear Dr. Ng,

Thank you very much for re-submitting your manuscript "Development and validation of a patient-level model to assess lifetime health outcomes of prediabetes and type 2 diabetes in Chinese populations (CHIME): a modelling study" (PMEDICINE-D-20-05652R3) for review by PLOS Medicine.

I have discussed the paper with my colleagues and the academic editor and it was also seen again by two reviewers. Provided the remaining editorial and production issues are dealt with, we are planning to accept the paper for publication in the journal.

[LINK]

We look forward to receiving the revised manuscript by Jun 11 2021 11:59PM.   

Sincerely,

Caitlin Moyer, Ph.D.

Associate Editor 

PLOS Medicine

plosmedicine.org

Requests from Editors:

1. From the Academic Editor: Please address the remaining comment pertaining to the issue of individual-level validation from Reviewer 3 as completely as possible. Validation of models plays an important role for quality assessment of this model. The authors have validated against real-world observational data and also from data reported in other references. As mentioned by reviewer 3, the model the authors propose can potentially be useful for population-level risk prediction of several endpoints considered together. If the issue of validation at the individual level cannot be objectively solved, please discuss this as a limitation and adjust the description throughout the manuscript (e.g. temper the emphasis on individual-level outcomes in the title, abstract, and main text).

2.Title: We suggest revising to: “Development and validation of the CHIME simulation model to assess lifetime health outcomes of prediabetes and type 2 diabetes in Chinese populations: A modelling study”

3. Author Summary: What did the researchers do and find? We suggest replacing “closer” with “stronger” or “more accurate” in the last bullet point, depending on the meaning intended.

4. Results: Line 382: Please clarify if this should read “...close correspondence with heart failure and ulcer of the skin.”

5. References: Ref 24, 30: Please abbreviate as “Lancet”

6. Figure 3: If possible, please choose colors for the plots that do not require distinguishing red/green as this may be difficult for some readers.

7. Supporting Information Files: For each of the Figures presented as supporting information files, please incorporate the title and legend into the file along with the figure.

8. S3 Figure: Please provide a descriptive title for each individual figure within the file.

9. S8 Table: Please provide a final version with the comment removed.

Comments from Reviewers:

Reviewer #1: The authors have addressed our previous comments.

Reviewer #3: I have the same comment as before: the Authors present the finding on collapsed data from trials (eg. prediction at population level) as a validation of finding at individual data level. I cannot really agree to accept this as a validation of their model. To be clear, the model the Authors propose can potentially be useful for population-level risk prediction of several endpoints considered together. This is not trivial but is simply different from trying to predict individual outcomes in a given patient.

[LINK]

---

## [Editor Report · Decision Letter 4]

11 Jun 2021

Dear Dr Ng, 

On behalf of my colleagues and the Academic Editor, Weiping Jia, I am pleased to inform you that we have agreed to publish your manuscript "Development and validation of the CHIME simulation model to assess lifetime health outcomes of prediabetes and type 2 diabetes in Chinese populations: A modelling study" (PMEDICINE-D-20-05652R4) in PLOS Medicine.

Along with completing the formatting requests, please submit a final version of S3 Figure, as the current PDF version contains comments.

PRESS

Sincerely, 

Caitlin Moyer, Ph.D. 

Associate Editor 

PLOS Medicine